# The Phylogeny, Metabolic Potentials, and Environmental Adaptation of an Anaerobe, *Abyssisolibacter* sp. M8S5, Isolated from Cold Seep Sediments of the South China Sea

**DOI:** 10.3390/microorganisms11092156

**Published:** 2023-08-25

**Authors:** Ying Liu, Songze Chen, Jiahua Wang, Baoying Shao, Jiasong Fang, Junwei Cao

**Affiliations:** 1Shanghai Engineering Research Center of Hadal Science and Technology, College of Marine Sciences, Shanghai Ocean University, Shanghai 201306, China; liuying06shengke@163.com (Y.L.); sjtu_wangjiahua@163.com (J.W.); shaobaoying0123@163.com (B.S.); 2The Guangxi Key Laboratory of Beibu Gulf Marine Biodiversity Conservation, College of Marine Sciences, Beibu Gulf University, Qinzhou 535000, China; 3Shenzhen Ecological and Environmental Monitoring Center of Guangdong Province, Shenzhen 518049, China; songzechen2012@sina.com; 4Laboratory for Marine Biology and Biotechnology, Qingdao National Laboratory for Marine Science and Technology, Qingdao 266000, China

**Keywords:** cold seep, *Bacillota*, *Abyssisolibacter* sp. M8S5, metabolic potentials, adaption genomic analysis

## Abstract

*Bacillota* are widely distributed in various environments, owing to their versatile metabolic capabilities and remarkable adaptation strategies. Recent studies reported that *Bacillota* species were highly enriched in cold seep sediments, but their metabolic capabilities, ecological functions, and adaption mechanisms in the cold seep habitats remained obscure. In this study, we conducted a systematic analysis of the complete genome of a novel *Bacillota* bacterium strain M8S5, which we isolated from cold seep sediments of the South China Sea at a depth of 1151 m. Phylogenetically, strain M8S5 was affiliated with the genus *Abyssisolibacter* within the phylum *Bacillota*. Metabolically, M8S5 is predicted to utilize various carbon and nitrogen sources, including chitin, cellulose, peptide/oligopeptide, amino acids, ethanolamine, and spermidine/putrescine. The pathways of histidine and proline biosynthesis were largely incomplete in strain M8S5, implying that its survival strictly depends on histidine- and proline-related organic matter enriched in the cold seep ecosystems. On the other hand, strain M8S5 contained the genes encoding a variety of extracellular peptidases, e.g., the S8, S11, and C25 families, suggesting its capabilities for extracellular protein degradation. Moreover, we identified a series of anaerobic respiratory genes, such as glycine reductase genes, in strain M8S5, which may allow it to survive in the anaerobic sediments of cold seep environments. Many genes associated with osmoprotectants (e.g., glycine betaine, proline, and trehalose), transporters, molecular chaperones, and reactive oxygen species-scavenging proteins as well as spore formation may contribute to its high-pressure and low-temperature adaptations. These findings regarding the versatile metabolic potentials and multiple adaptation strategies of strain M8S5 will expand our understanding of the *Bacillota* species in cold seep sediments and their potential roles in the biogeochemical cycling of deep marine ecosystems.

## 1. Introduction

Cold seeps, which are widely distributed on the seafloor of the world’s oceans, play a crucial role in the global marine geochemical cycling of carbon and sulfur [1,2,3]. Cold seep environments are characterized by methane-rich fluid emissions and distinctive sulfur oxidation–reduction reactions and support seabed oasis ecosystems composed of various kinds of high-productive microbial and faunal assemblages [2,4,5,6,7]. Therefore, diverse and abundant organic matter exists in cold seep environments, including gaseous and liquid hydrocarbons, aromatic compounds, polysaccharides, and biomass from chemosynthetic microorganisms and invertebrates [3,8,9,10,11,12]. However, much less is known about which microorganisms are capable of degrading the organic matter and how these degradation processes happen in cold seeps.

The phylum *Bacillota* is predominantly low G+C content, gram-positive, rod-shaped, obligate anaerobic bacteria capable of forming spores to resist extreme environmental conditions [13,14,15]. *Bacillota* is one of the most abundant bacterial groups and is widely distributed in a variety of environments, including cold seeps [5,16], marine environments [17,18,19,20,21], soils [22,23,24,25], oil fields [15,26], freshwater lakes [27], the human gut [28], plateau anaerobic batch reactors [29], and the rhizosphere of plants [30]. Notably, Cui et al. (2019) found that *Bacillota* was the dominant bacterial group in the cold seep sediments of the South China Sea (SCS), with up to 37% of the relative abundance in the total bacterial communities [5]. Metabolically, many members of *Bacillota* play an essential role in the biodegradation of lignocellulosic and carbohydrate polymers [23], polychlorinated biphenyl [29], and petroleum hydrocarbons [31]. Other studies have shown that Clostridia (assigned to the phylum *Bacillota*) has the capability to utilize various complex carbohydrates, such as starch, cellulose, chitin, and xylan, and the *Bacillus* sp. of this phylum may play an important role in long-chain alkane biodegradation [32,33]. Together, these findings suggest that *Bacillota* plays a crucial role in biogeochemical cycles.

The genus *Abyssisolibacter* belongs to the order Clostridiales and the family Clostridiaceae within the phylum *Bacillota*. At present, the genus *Abyssisolibacter* contains only one isolated strain, *Abyssisolibacter fermentans* MCWD3, which was isolated from deep-sea sediment (depth 2150 m) obtained from the Ulleung Basin, East Sea, Korea [34]. Strain MCWD3 was a gram-staining-negative and anaerobic bacterium and could utilize fructose, mannose, cellobiose, and methionine only in the presence of yeast extract [34]. Therefore, the characteristics of the substrate requirements of strain MCWD3 remain largely uncertain, and little is known about the metabolic potential and ecological functions of the species in the genus *Abyssisolibacter*.

Hitherto, more than 40 cold seep sites have been detected in the South China Sea (SCS), only two of which are active: the Haima cold seep and the Site F (or Formosa Ridge) cold seep [2,4,12,35,36,37]. The cold seep at Site F was found in 2007 and is located at a depth of 1120 m on the Formosa Ridge of the northeastern SCS, offshore of southwestern Taiwan [36,38,39,40]. This cold seep is enriched by chemosynthetic communities, such as deep-sea mussels and galatheid crabs, which may provide various types of organic matter to support the growth of heterotrophic microbial communities [41]. In this study, we report a novel anaerobic strain, *Abyssisolibacter* sp. M8S5, which was isolated from the cold seep sediments of Site F at a depth of 1151 m. Our analysis of the complete genome sequence of strain M8S5 revealed the phylogeny, metabolic versatility, and multiple adaptive strategies of M8S5 in the high-pressure, low-temperature, and anaerobic cold seep environment.

## 2. Materials and Methods

### 2.1. Sample Description and Strain Isolation

Remotely operated vehicle (ROV) sediment push cores were collected at Site F (22°07′ N, 119°17′ E) at a depth of 1151 m from the cold seep of the South China Sea (Appendix A) in May 2018. After recovering the cores onboard, the sediment samples were anaerobically preserved in sterilized seawater at 4 °C.

Once in the lab, a subsample was used to inoculate M89 medium, prepared with a gas phase of N_2_ (200 kPa), and incubated anaerobically at 35 °C. The M89 medium contained (L^−1^) 700 mL seawater, 300 mL distilled water, 1 g tryptone, 2 g yeast extract, 2 g casamino acid, 2 g 4-aminobutyric acid, 3 g HEPES buffer, 0.25 g l-cysteine hydrochloride dihydrate (as a reducing agent), and 1 mg sodium resazurin (as a redox indicator). The pH of the medium was adjusted to 7.0 with 5 M NaOH at room temperature. After autoclaving, 1 mL of vitamin solution [42] and 1 mL of selenite–tungstate solution [43] were added. After 10 days of incubation, the enrichments were sub-cultured under the same conditions and purified by eight repeated dilutions-to-extinction series. One isolate, designated as M8S5, was obtained. Stock cultures were stored at −80 °C with 5% (*v*/*v*) DMSO.

### 2.2. DNA Extraction, Genome Sequencing, and Assembly

The cells of strain M8S5 were cultured in M89 medium at 35 °C for 5 days. The genomic DNA of strain M8S5 was extracted using the Qiagen DNA extraction kit (Qiagen, Hilden, Germany). The total DNA was used for quality control and was obtained by gel electrophoresis and further quantified by a Qubit fluorometer (Thermo Fisher Scientific, Waltham, MA, USA).

The complete genome sequence of strain M8S5 was sequenced using Oxford Nanopore Sequencing Technologies at Nextomics Biosciences Co., Ltd. (Wuhan, China). Oxford Nanopore Technologies sequencing libraries were prepared using the manufacturer’s ligation sequencing kit (SQK-LSK109), and sequencing was carried out on a PromethION device using flow cells (FLO-PRO002) without fragmentation. Finally, Nanopore reads were assembled using Canu (version 1.7) [44] in the normal assembly mode.

### 2.3. Gene Annotation and Metabolic Pathway Reconstruction

Genome annotation and ORF prediction in M8S5 and the other four strains were performed by the NCBI prokaryotic genome annotation pipeline (PGAP) [45]. The whole genome map of strain M8S5 was generated using the Circos visualization tool [46]. Metabolic pathway analysis of M8S5 and the other four related strains was performed by searching the KEGG GENES database [47] using BlastKOALA [48]. The gene functional annotation of M8S5 and MCWD3 was predicted by a BLASTP search in the Clusters of Orthologous Groups of proteins (COGs) database [49]. Carbohydrate hydrolases were identified in the genomes of M8S5 and the other four strains based on BLASTP searches against the CAZy database (Carbohydrate-Active Enzymes database) [50]. The number and size of the genomic islands in strain M8S5 were determined using the IslandViewer 4 server [51]. The transporters of M8S5 and the other four strains were determined by BLASTP against TransportDB 2.0 [52]. The peptidases of M8S5 and the other four strains were predicted by BLASTP against MEROPS [53]. The signal peptides of M8S5 and the other four strains were predicted by SignalP v.5.0 [54].

### 2.4. Phylogenetic Tree Construction

The taxonomy annotation of strain M8S5 and the related species was performed using the Genome Taxonomy Database Toolkit (GTDB-Tk, version 3) [55] with the Genome Taxonomy Database (GTDB, release R08-RS214). Then, these protein sequences were separately aligned using Clustal Omega [56], and all the aligned sequences were manually degapped and tandemly connected. The phylogenetic tree was constructed using FastTree 2 [57] with the neighbor-joining method, and the robustness of the tree was evaluated by bootstrap analysis with 1000 replicates. Finally, the phylogenetic tree based on 120 bacterial conserved marker genes generated by GTDB-Tk was visualized using iTOL [58].

### 2.5. Comparative Genomic Analyses

The ribosomal RNAs (rRNAs) of strain M8S5 and the other four strains were predicted using Barrnap version 0.9 with the default parameters [59]. Then, the 16S rRNA gene sequences of the five strains were extracted and compared with each other using BLASTn [60]. Finally, the values of 16S rRNA gene similarity between strain M8S5 and the other four species were obtained.

The protein sequence alignment was conducted by BLASTp with the following cut-off values: identity, 50%; query coverage, 50%; E-value, 1 × 10^−10^. The proteins employed by strain M8S5 but not by strain MCWD3 were considered to be strain-specific. On the other hand, the proteins shared by strain M8S5 and MCWD3 but absent in the other three species (*Senegalia massiliensis* SIT17, *Senegalia massiliensis* 583, and *Clostridiisalibacter paucivorans* DSM 22131) were considered to be genus-specific.

The average nucleotide identity (ANI) is referred to as the mean of the identity values between homologous genomic regions shared by two genomes [61]. The ANI is used to compare the closely related genomes of strains [62]. The ANI was calculated between M8S5 and the other four species using the JSpeciesWS server with the BLAST algorithm [63].

The average amino acid identity (AAI) can provide better resolution when comparing highly different bacterial species, such as species from different genera [62,64]. The AAI was estimated between M8S5 and the other four strains using compareM v0.1.0.

### 2.6. 16S rRNA gene Distribution Analysis of Strain M8S5

The full-length 16S rRNA gene sequences of strain M8S5 were used to screen for the presence of closely related sequences in all publicly available Short Read Archive (SRA, www.ncbi.nlm.nih.gov/sra, accessed on 1 May 2023) datasets with Integrated Microbial Next Generation Sequencing (IMNGS, 99% identity threshold and size over 200 nucleotides) [65], the metadata for the matched samples was obtained from NCBI, and the latitude/longitude coordinates were plotted using Ocean Data View [66].

## 3. Results and Discussion

### 3.1. The Genomic Features of Strain M8S5

The complete genome of strain M8S5 contains one single chromosome with a total length of 4,683,267 base pairs (bp), which is slightly smaller than the scaffold genome of *Abyssisolibacter fermentans* MCWD3 (4,789,356 bp). The G+C content of the M8S5 genome is 30.3%. The genome of strain M8S5 consists of 4376 genes, including 4250 protein-coding genes, 7 rRNA operons (4 with the pattern of 16S-23S-5S, and 3 with the pattern of 16S-tRNA^Ala^-23S-5S), 77 tRNAs, 4 ncRNAs, and 24 pseudogenes (Table 1).

For COG classification, about 1575 (35.9%) and 1729 (40.8%) protein-coding genes of strains M8S5 and MCWD3, respectively, were assigned to 23 different clusters of orthologous groups (COGs) categories (Appendix A). Such a low percentage of COG annotation in the two strains may indicate that the genome of *Abyssisolibacter* species contains a high proportion of novel genes and may be valuable for further studies. The top five COG categories in strain M8S5 included translation, ribosomal structure and biogenesis (COG-J, 11.14%), amino acid transport and metabolism (COG-E, 10.91%), transcription (COG-K, 9.02%), general function prediction only (COG-R, 7.48%), and energy production and conversion (COG-C, 6.68%) (Appendix A). A graphical map of the complete genome of strain M8S5 is shown in Figure 1.

The proteins employed by strain M8S5 but not by strain MCWD3 were considered to be strain-specific. On the other hand, the proteins shared by strain M8S5 and MCWD3 but absent in the other three species (*Senegalia massiliensis* SIT17, *Senegalia massiliensis* 583, and *Clostridiisalibacter paucivorans* DSM 22131) were considered to be genus-specific. As many as 20 genomic islands (GIs) were predicted in strain M8S5, and, in particular, seven of these GIs were related to nitrogenase and endospore germination permease, suggesting that this genome harbors obvious evidence of horizontal gene transfer (Appendix A). Moreover, the genes encoding six transposases, nine recombinases, one site-specific tyrosine recombinase, and three tyrosine-type recombinases/integrases were identified in strain M8S5, and more than half of the enzymes (10, 52.6%) were specific in strain M8S5 (Appendix A).

### 3.2. The Phylogeny of Strain M8S5

The alignment of the 16S rRNA gene sequences revealed that strain M8S5 was closely related to *Abyssisolibacter fermentans* MCWD3, followed by *Senegalia massiliensis* SIT17, *Senegalia massiliensis* 583, and *Clostridiisalibacter paucivorans* DSM 22131, with identities of 96.87%, 94.75%, 91.92%, and 89.74%, respectively (Appendix A). Nevertheless, most of the species in the phylum *Bacillota* have 16S rRNA gene sequence heterogeneity [67], and the seven 16S rRNA gene sequences in strain M8S5 were different from each other (with the identity lower than 100%). Therefore, phylogenetic analysis based on the 16S rRNA gene may result in biases in the phylogeny of strain M8S5.

In this study, we constructed a phylogenetic tree based on 120 conserved protein sequences (known as GTDB taxonomy) (Figure 2). It showed that strain M8S5 is affiliated with the genus *Abyssisolibacter* and is closely related to *Abyssisolibacter fermentans* MCWD3 (Figure 2), which was isolated from deep-sea sediments of the Ulleung Basin, East Sea, Korea [34].

In addition, the average amino acid identity (AAI) evaluates genome-wide identity between distance species. Our results showed that the AAI values between M8S5 and the related four species ranged from 57.27 to 69.97%, and the best-matched strain was *A. fermentans* MCWD3 (Appendix A). The average nucleotide identity (ANI) estimates genome-wide identity between closely related species. Our results revealed that the ANI value between strain M8S5 and *A. fermentans* MCWD3 was 77.85% (Appendix A), suggesting the closest evolution distance between the two strains. The ANI values between M8S5 and the other three species could not be calculated due to the relatively distant evolution relationships. These pieces of evidence indicated that strain M8S5 is affiliated with the genus *Abyssisolibacter*, the family Clostridiaceae, and the order Clostridiales within the phylum *Bacillota*.

### 3.3. The Environmental Distribution of Abyssisolibacter Species

To study the environmental distribution of this genus with very few isolates, we searched and collected a total of 16 16S rDNA amplicons, which showed >99% identity with strain M8S5 (Appendix A). The possible environmental distribution of strain *Abyssisolibacter* M8S5 is shown in Figure 3. It shows that *Abyssisolibacter* species are widely distributed in various marine environments, including seawater (18 of the sample locations), marine sediments (10), coral reefs (9), sponges (7), and an oyster gut (1) (dots with blue color in Appendix A). Therefore, we propose that the genus *Abyssisolibacter* may play an important role in biogeochemical cycling in marine ecosystems.

### 3.4. Metabolic Characteristics of Strain M8S5

Cold seep habitats possess diverse abundant organic matter, such as methane, petroleum, microorganisms, and macrofauna [1]. Previous studies showed that *Bacillota* could utilize various kinds of organic matter [68]. To study the metabolic characteristics and possible ecological functions of strain M8S5 in cold seep habitats, the metabolic pathways of strain M8S5 were reconstructed and compared to those of four neighboring strains (Figure 4).

(1)Carbohydrate metabolism

Metabolic analysis showed that strain M8S5 has the potential to utilize various kinds of carbohydrates, including chitin, cellulose, trehalose, arabinogalactan oligomer/maltooligosaccharide, maltose, sucrose, mannose, glucose, sugar, cellobiose, and galactitol (Figure 4 and Appendix A). Among them, the genes encoding the sucrose PTS system (*scrA*) and mannose (*manA*) were detected in the two isolated *Abyssisolibacter* strains (M8S5 and MCWD3) but not in the three neighboring species (*S. massiliensis* SIT17, *S. massiliensis* 583, and *C. paucivorans* DSM 22131). Moreover, strain M8S5 harbored eight genes encoding fructose transporter, including a strain-specific gene (*fruAb*) (Figure 4 and Appendix A). These results suggest that sucrose and fructose might be important carbon sources for strain M8S5. In addition, the genes encoding maltose (*mapA*), trehalose transporter (*treBC* and *malX*), and arabinogalactan oligomer/maltooligosaccharide transporters (*ganOPQ*) were found in strain M8S5 but not in strain MCWD3, implying that the utilization of these sugars may contribute to the survival of strain M8S5 to the in situ cold seep environment.

Chitin is a polysaccharide that exists in macrofauna, including marine shellfish, which are abundant in cold seep environments [1,12,38]. Chitinase (EC:3.2.1.14) could decompose chitin to chitobiose or GlcNAc. Five genes encoding chitinase were detected in strain M8S5 (Figure 4 and Appendix A). Moreover, the complete pathways of cellulose degraded to cellodextrin (EC:3.2.1.4), cellobiose (EC:3.2.1.21 and EC:3.2.1.4), and glucose (EC:3.2.1.21) were identified in strain M8S5 but not in strain MCWD3 (Appendix A), suggesting a strain-specific ecological function in local habitats. In addition, phosphate acetyltransferase (EC:2.3.1.8) and acetate kinase (EC:2.7.2.1) were predicted in both *Abyssisolibacter* strains, suggesting they could alleviate the accumulation of aceyl-coA generated from glycolysis via acetate production and export (Figure 4). Taken together, these findings highlight the ecological potentials of *Abyssisolibacter* species in the organic biogeochemical cycling of carbohydrates, especially insoluble oligosaccharides, in cold seeps.

(2)Amino acid metabolism and extracellular protein degradation

Regarding the biosynthesis of amino acids, our genomic survey showed that the strain M8S5 preoccupies the complete pathways for the biosynthesis of 18 kinds of amino acids, whereas the biosynthesis pathways of histidine and proline were largely incomplete (Figure 4 and Appendix A). Similarly, strain MCWD3 could also disable to biosynthesize histidine. The absence of these amino acid synthetic genes indicates that the strains M8S5 and MCWD3 are dependent on the intake of histidine and proline, which may originate from the mussels, clams, and tubeworms that thrive in cold seep environments [1,12,32,38].

On the other hand, strain M8S5 possesses a larger number of genes encoding amino acid and peptide transporters, which may help the intake of peptide (*ddpDF*), oligopeptide (*oppABCDF*), arginine/lysine/histidine (*artQPR*), alanine, proline/glycine (*opuA*), glutamate/aspartate, and polar amino acid from the extracellular cold seep environments (Figure 4 and Appendix A). Moreover, considering that the prosperity of macrofauna (such as mussels, clams, and tubeworms) in the cold seep contains abundant proteins [1,12,38], we propose that the secretion of extracellular peptidases is an important physiological characteristic of M8S5. In this study, approximately 28 different genes of SignalP-fused peptidases belonging to 17 peptidase subfamilies (C25, C39, M1, M4, M12, M23B, M28, M41, M50B, S8A, S8B, S9B, S11, S12, S41A, S41B, and S49C) were predicted in M8S5, among which S8 peptidase was found to be remarkably abundant (Appendix A). The S8 family, generally consisting of the S8A (subtilisin) and S8B (kexin) subfamilies, is the second largest family of serine-type peptidases, which has a catalytic triad containing aspartate, histidine, and serine. Most members of the S8 peptidase family are secreted and non-specific and may be involved in nutrition intake and further meet the protein requirements of microorganisms [69]. A total of seven extracellular S8 family peptidases were detected, six of which were genus-specific, and one was strain-specific (Appendix A). Interestingly, three genes of the S8A family peptidase form an operon in strains M8S5 and MCWD3 (Appendix A). Such organization was previously observed in several *Kangiella* strains, which were proposed as extraordinary extracellular protein degraders [70]. These findings suggest that the secretion of extracellular proteases may be an important survival strategy for *Abyssisolibacter* species.

(3)d-amino acid metabolism

It is well known that bacteria produce diverse d-amino acids (DAAs), which are essential components of bacterial cell wall peptidoglycan and are also found in various non-ribosomal peptides produced by microbes [71,72,73]. A high proportion of DAAs may indicate that abundant refractory organic matter exists in the cold seep environment [74].

In this study, 21 genes involved in d-amino acid transport and utilization were identified in strain M8S5 (Figure 4 and Appendix A), including d-methionine transport (*metNIQ*), d-cysteine desulfhydrase (*dcyD*), d-proline reductase (*prdABDE*), and d-alanine hydrolase (*fmtA*) as well as four racemases of amino acids, which catalyze the conversion between l- and d-amino acids (including alanine racemase [EC:5.1.1.1], glutamate racemase [EC:5.1.1.3], ornithine racemase [EC:5.1.1.12], and aspartate racemase [EC:5.1.1.13]). These pieces of evidence suggest that strain M8S5 might contribute to the degradation of d-amino acid-containing recalcitrant organic polymers in cold seep environments.

(4)Nitrogen metabolism

Cold seep ecosystems are mainly supported by anaerobic methanotrophy, and the abundant input of methane (a non-nitrogenous carbon source) may drive the cold seep microbial community to nitrogen limitation [75,76]. In this study, various kinds of transporters and metabolic pathways of nitrogen sources were predicted in strain M8S5, including ethanolamine, spermidine, putrescine, nitrogenase, and ammonia (Figure 4 and Appendix A).

Ethanolamine (EA), derived from the membrane phospholipid phosphatidyl-ethanolamine of bacterial cells, can be used as a valuable source of carbon and/or nitrogen by a variety of bacteria (including *Bacillota*, *Flavobacterium*, and *Salmonella*) from the environments [77,78,79]. In the present study, 17 genes encoding ethanolamine transporter (encoding by *eutH*) and utilization (encoding by *eutABCJLMNPQS*) were predicted in strain M8S5 (Figure 4 and Appendix A). Among these genes, *eutB* and *eutC* are the central ethanolamine utilization genes, which encode the large and small subunits of ethanolamine ammonia-lyase (EC:4.3.1.7), respectively [78]. Intriguingly, only strain M8S5 possessed the genes of *eutB* and *eutC*, which catabolize ethanolamine into ammonia and acetaldehyde by an ethanolamine ammonia lyase (Figure 4 and Appendix A). Therefore, these findings suggest that the utilization of ethanolamine may be an important metabolic characteristic for strain M8S5 to survive in cold seep ecosystems.

Polyamine compounds, including spermidine and putrescine, are a group of ubiquitous polycations necessary for cell growth and play essential roles in the biosynthesis of nucleic acids and proteins in microorganisms [80,81,82]. Moreover, recent studies reported that diverse kinds of epifauna thrived in the cold seep environment of the South China Sea [38], which may produce a large number of spermidines and putrescines. Our results showed that a total of nine genes encoding spermidine and putrescine transporters (*potABCD*) were identified in strain M8S5 (Figure 4 and Appendix A). Moreover, the enzymes of spermidine and putrescine degradation were also predicted, which finally transform polyamine compounds into ammonia (Figure 4). Among the transport genes of spermidine and putrescine, four were found to be absent in strain MCWD3, indicating that M8S5 may gain more genes via horizontal gene transfer (HGT) for the utilization of the polyamine compounds enriched in cold seep environments.

Biological nitrogen fixation (referred to as diazotrophy), which is the reduction of atmospheric dinitrogen to ammonia, is an important way to obtain a nitrogen source for microorganisms [83]. Previous studies reported that members of *Bacillota* were detected as diazotrophs, which contributed to the nitrogen balance of deep-sea sediments [84]. In our study, we identified the complete pathways of nitrogen fixation (catalyzed by nitrogenase and encoded by *nifEH*) in strain M8S5, but these were absent in the four phylogenetically related strains: MCWD3, *S. massiliensis* SIT17, *S. massiliensis* 583, and *C. paucivorans* DSM 22131 (Figure 4 and Appendix A). These findings may imply that strain M8S5 has gained the capability to transform inorganic nitrogen to organic nitrogen via horizontal gene transfer (HGT) to survive in nitrogen-limited cold seep environments.

(5)Nucleic acid metabolism

For nucleic acid metabolism, four genes encoding nucleoside transporters were identified in strain M8S5 (Figure 4 and Appendix A), and the biosynthetic pathways of purine and pyrimidine were complete. However, the degradation pathways of nucleotides were absent in strain M8S5, indicating that these nucleotides may constitute DNA, RNA, and other cellular components but may not be utilized as carbon and energy sources by strain M8S5.

(6)Glycine reductase in substrate-level phosphorylation

Glycine reductase (EC:1.21.4.2) is present in many obligate anaerobic microbes (e.g., the *Bacillota* species *Eubacterium acidaminophilum*) that could reduce glycine (acts as an electron acceptor) to acetyl-phosphate in Stickland reactions, with thioredoxin (acts as an electron donor) transformed to thioredoxin disulfide, which finally generates ATP [85,86].

In this study, we identified two gene operons that encoded glycine reductase and thioredoxin systems in strain M8S5 (Figure 5 and Appendix A). We found that one gene operon encoded betaine reductase (*grdI* and *grdH*), thioredoxin 1 (*trxA*), glycine/sarcosine/betaine reductase (*grdA*), glycine/betaine transporter (*betL*), glycine/sarcosine/betaine reductase (*grdC* and *grdD*), and NADPH-dependent thioredoxin reductase (*trxB*) (Figure 5A and Appendix A). The other gene operon encoded GrdX family protein (*grdX*), NADPH-dependent thioredoxin reductase (*trxB*), thioredoxin 1 (*trxA*), and glycine reductase (*grdE*, *grdA*, *grdB*, *grdC*, and *grdD*) (Figure 5B and Appendix A). These genes could conserve energy via the promotion of substrate-level phosphorylation and allow strain M8S5 to adapt well to the anaerobic and organic matter-enriched cold seep habitats.

### 3.5. Genes Involved in High-Pressure and Low-Temperature Adaptation

To uncover how strain M8S5 adapted to the cold seep environments, we analyzed its genome and identified a large number of genes that were potentially involved in high hydrostatic pressure (HHP) and low-temperature adaptation (Appendix A), including the genes in osmoprotectants, molecular chaperones, antioxidation, and sporulation.

Compatible solutes, commonly referred to as osmoprotectants, are low molecular weight, highly soluble organic compounds that mainly protect microorganisms from a variety of environmental stresses, such as drought, high salt, low or high temperature, HHP, and heavy metals [87,88,89,90,91,92,93]. In this study, we found that strain M8S5 possesses 21 genes to transport diverse kinds of compatible solutes, including glycine betaine/carnitine/choline, proline, sarcosine, and trehalose (Figure 4 and Appendix A). Glycine betaine is widespread in a variety of microorganisms and is thought to primarily serve as a potent osmoprotectant that may stabilize proteins and membrane structures, maintain cell turgor and volume, and act as a cryoprotectant under extreme environments [87,88,89,91,94]. Strain M8S5 possessed glycine/betaine/proline (encoding by *opuABCD*) and three BCCT family transporters (Figure 4 and Appendix A). Likewise, trehalose, which also acts as an important osmoprotectant, may stabilize membranes at low temperatures, protect proteins from thermal denaturation, enhance resistance to hypoxia, and serve as a source of energy [87,88,94,95]. A strain-specific gene (*treB*) encoding trehalose transporter was also detected in strain M8S5 (Figure 4 and Appendix A). These findings suggest that the intake of osmoprotectants might be a crucial strategy for M8S5 to adapt to local cold seep environments.

Moreover, a total of five genes encoding molecular chaperones, including *dnaJK*, *grpE*, *htpG*, and *hslO*, were predicted in strain M8S5 (Appendix A). These molecular chaperones may be induced by high-pressure and low-temperature stresses, help proteins to fold correctly, and maintain protein function [87,92,95]. Also, two genes related to cold shock proteins (*cspA*) were detected in the genome of strain M8S5 (Appendix A), which may serve as cryoprotectants in low-temperature adaptation [96]. In addition, HHP and low temperatures also produce reactive oxygen species (ROS), which are harmful to anaerobic microbes [97]. We identified three peroxidases and one superoxide dismutase-related gene in strain M8S5 (Appendix A), which may serve as antioxidants and help to scavenge intracellular ROS and, therefore, protect the cells from cold and/or HHP stress.

Finally, spores are metabolically dormant and highly resistant structures formed by *Bacillota* that enhance their survival in adverse environmental conditions, such as HHP and low temperatures. Spore formation is essential to the survival of some obligate anaerobic *Bacillota* bacteria [98,99]. In this study, a total of 69 spore-related genes were identified in strain M8S5, mainly including spore germination protein (encoding by *gerKABC*, *gerM*, *yaaH*, *yndDE*, and *ypeB*), spore maturation protein (*cgeBD* and *spmAB*), sporulation protein (encoding by *spoIIABDEGMPR*, *spoIIIAABCDEFGH*, *spoIVABF*, and *spoVABCDEGS*, respectively), and spore coat protein (*cotAEHIJS*) (Appendix A). These findings may indicate that sporulation may help strain M8S5 to adapt to many extremely unfavorable environments.

## 4. Conclusions

In this study, we report the complete genome sequence of a novel anaerobic bacterial strain, M8S5, which was isolated from the Site F cold seep in the South China Sea at a depth of 1151 m. Phylogenetically, strain M8S5 belongs to the genus *Abyssisolibacter* and the phylum *Bacillota*. Metabolically, strain M8S5 could potentially utilize various kinds of simple and complex polysaccharides, such as chitin, cellulose, and trehalose. Although strain M8S5 lacks genes for synthesizing histidine and proline, it possesses not only the transporter genes of many amino acids and peptides but also diverse genes of extracellular peptidases, implying that it is capable of degrading extracellular proteins in the cold seep. Furthermore, the strain can utilize a variety of organic and inorganic nitrogen sources, such as ethanolamine, spermidine, putrescine, nitrogen, and ammonia. The genes of diverse transporters of osmoprotectants, molecular chaperones, and reactive oxygen species-scavenging proteins as well as spore formation may help it to adapt well to high-pressure and low-temperature cold seep environments. The findings regarding the versatile metabolic potentials and adaptation strategies of strain M8S5 provide new insights into the contribution of *Abyssisolibacter* species to biogeochemical cycling and expand our understanding of *Bacillota* species in cold seep habitats.

## Figures and Tables

**Figure 1 microorganisms-11-02156-f001:**
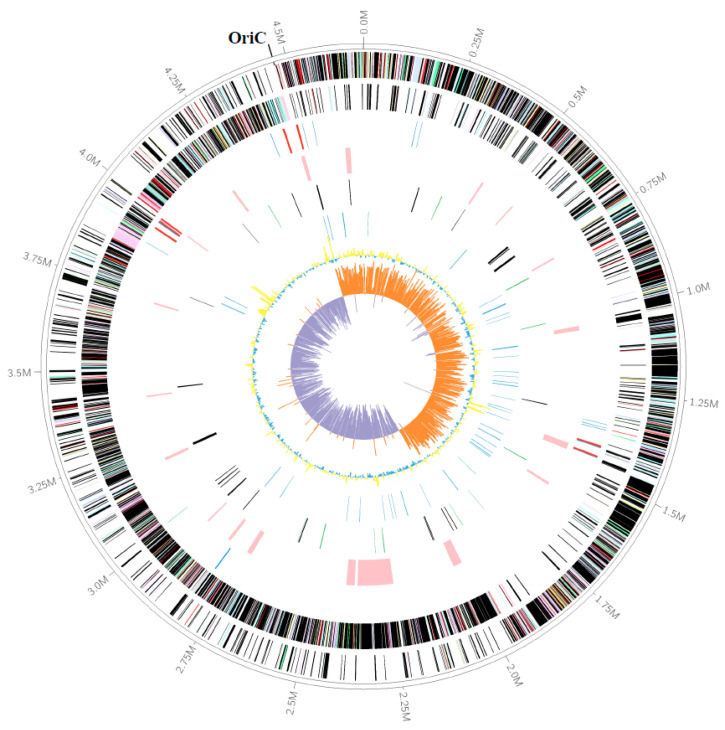
Graphical representation of M8S5 genome. Genes on the forward (shown in the outer circle) and reverse (shown in the inner circle) strands are colored according to their cluster of orthologous gene (COG) categories (except those colored in black for no hits); tRNA and rRNA genes are highlighted with blue and red colors, respectively; gene islands are shown in green; genes encoding D-amino acids and peptidases are shown in green and orange, respectively; genes encoding polysaccharides are shown in blue color; GC content is shown in yellow and blue; and GC skew is shown in orange and purple (window, 5000 bp; step, 2500 bp).

**Figure 2 microorganisms-11-02156-f002:**
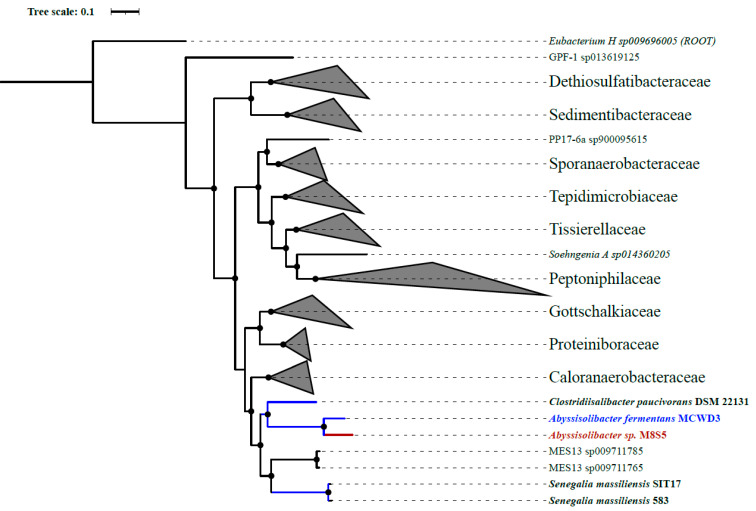
The phylogenetic tree of strain M8S5 was constructed based on 120 concentration proteins. Strains M8S5 and MCWD3 are highlighted in red and blue color, respectively. The black nodes represent the bootstrap values ≥ 80.

**Figure 3 microorganisms-11-02156-f003:**
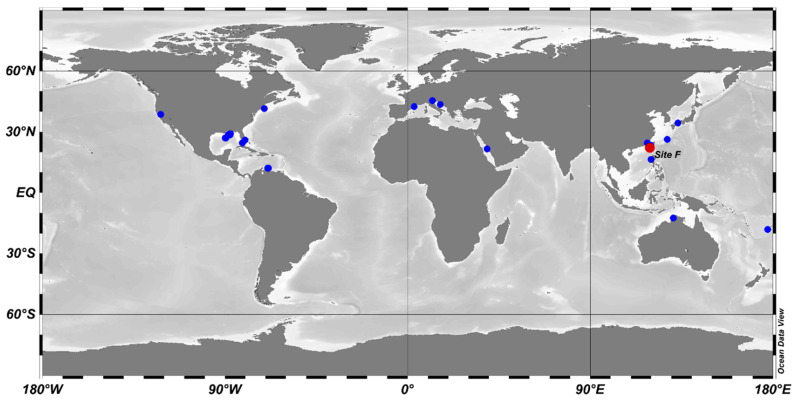
Environmental distribution of the strain M8S5. The sampling site F in this study is highlighted in red color.

**Figure 4 microorganisms-11-02156-f004:**
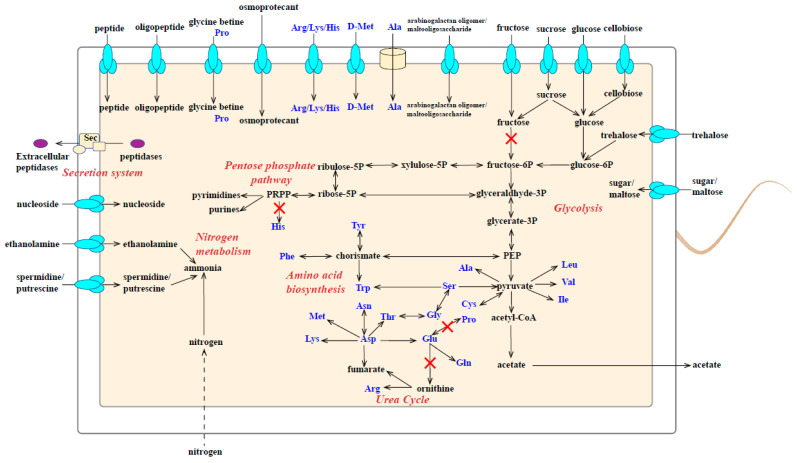
Reconstructed metabolic pathways of strain M8S5. X marks in the red color represent the genes absent in M8S5.

**Figure 5 microorganisms-11-02156-f005:**
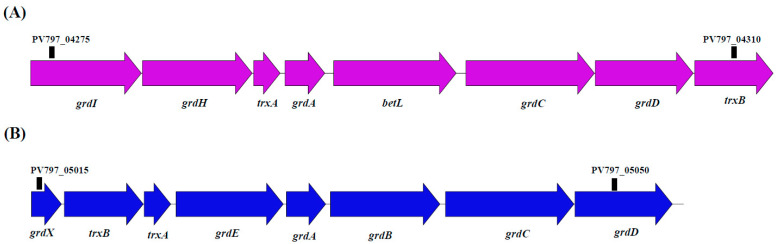
Two gene clusters of type glycine reductase in the genome of strain M8S5. (**A**) glycine reductase I, (**B**) glycine reductase II.

**Table 1 microorganisms-11-02156-t001:** Genome features of strain M8S5.

Items	M8S5
Size (bp)	4,683,267
G+C content (%)	30.3
Coding sequence (%)	83.76
Total genes	4376
Protein-coding genes	4250
Pseudo genes	24
Genes assigned to COG	1575
rRNA operons	7
tRNA genes	77
ncRNA genes	4
Gene islands	20

## Data Availability

The complete genome sequence of *Abyssisolibacter* sp. M8S5 is available in GenBank under accession number CP118523.1.

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
