# Peer review of "The Phylogeny, Metabolic Potentials, and Environmental Adaptation of an Anaerobe, Abyssisolibacter sp. M8S5, Isolated from Cold Seep Sediments of the South China Sea"

_microorganisms, 2023, doi:10.3390/microorganisms11092156_

Round 1
Reviewer 1 Report
Manuscript is dedicated to the description and analysis of phylogenetic, metabolic and adaptation features of a novel anaerobic strain Abyssisolibacter sp. M8S5. The study is well designed and performed, the text is well written and comprehensive. The interpretation of the genome analysis is extensive and insightful. The only one serious remark is related to the lack of the genome sequenced on Illumina in addition to the Nanopore genome sequencing. It is worth combining these to technologies for more accurate genome base calling. However, it is optional. The minor remarks and corrections are in the file attached below.

Reviewer 2 Report
The paper by Liu et al shows that the novel anaerobic bacterial strain M8S5, belonging to the genus Abyssisolibacter and phylum Bacillota, has versatile metabolic potentials and multiple adaptation strategies that allow it to survive in the cold-seep environments. The strain can potentially utilize various kinds of simple and complex polysaccharides, amino acids, and nitrogen sources. The findings of the study provide new insights into the contribution of Abyssisolibacter species to the biogeochemical cycling and expand our understanding of the Bacillota species in the cold-seep habitats.
Overall, study should be of broad interest to readers of deep sea biogeochemical processes and anerobic respiration. The work is well designed and executed. Figure 1 is quite dense and can't be interpreted in a useful way. I suggest a revision here.
